# Telomere repeats induce domains of H3K27 methylation in Neurospora

Kirsty Jamieson[1†], Kevin J McNaught[1†], Tereza Ormsby[1], Neena A Leggett[1], Shinji Honda[2], Eric U Selker[1]*

[1]Institute of Molecular Biology, University of Oregon, Eugene, United States; [2]Faculty of Medical Sciences, University of Fukui, Fukui, Japan

**Abstract** Development in higher organisms requires selective gene silencing, directed in part by di-/trimethylation of lysine 27 on histone H3 (H3K27me2/3). Knowledge of the cues that control formation of such repressive Polycomb domains is extremely limited. We exploited natural and engineered chromosomal rearrangements in the fungus *Neurospora crassa* to elucidate the control of H3K27me2/3. Analyses of H3K27me2/3 in strains bearing chromosomal rearrangements revealed both position-dependent and position-independent facultative heterochromatin. We found that proximity to chromosome ends is necessary to maintain, and sufficient to induce, transcriptionally repressive, subtelomeric H3K27me2/3. We ascertained that such telomere-proximal facultative heterochromatin requires native telomere repeats and found that a short array of ectopic telomere repeats, $(TTAGGG)_{17}$, can induce a large domain (~225 kb) of H3K27me2/3. This provides an example of a *cis*-acting sequence that directs H3K27 methylation. Our findings provide new insight into the relationship between genome organization and control of heterochromatin formation.

DOI: https://doi.org/10.7554/eLife.31216.001

*For correspondence:
selker@uoregon.edu

†These authors contributed equally to this work

Competing interests: The authors declare that no competing interests exist.

## Introduction

Methylation of lysine 27 on histone H3 (H3K27me) has emerged as an important repressive mark of the Polycomb group (PcG) system, which is critical for development in higher organisms. PcG proteins were initially discovered in *Drosophila melanogaster* as repressors of homeotic (*HOX*) genes during early embryogenesis (*Lewis, 1978*) and play integral roles in the maintenance of cellular identity and differentiation in a variety of eukaryotes. Moreover, dysfunction of the PcG system commonly leads to disease, including cancer (*Piunti and Shilatifard, 2016*; *Conway et al., 2015*). Biochemical work demonstrated that PcG proteins form two distinct histone-modifying complexes known as Polycomb Repressive Complex 1 and 2 (PRC1 and PRC2) (*Müller et al., 2002*; *Wang et al., 2004*). PRC1 mono-ubiquitinates lysine 119 on histone H2A (H2AK119ub1) with its E3-ubiquitin ligase subunit, Ring1, while PRC2 catalyzes mono-, di-, and trimethylation of histone H3 lysine 27 (H3K27me1/2/3) by its SET-domain component, EZH2 (*Müller et al., 2002*; *Wang et al., 2004*). PRC2, but not PRC1, is widely conserved in eukaryotes, including the filamentous fungus *Neurospora crassa*, but is absent in some simple eukaryotes such as the well-studied yeasts *Saccharomyces cerevisiae* and *Schizosaccharomyces pombe* (*Jamieson et al., 2013*). H3K27me2/3 covers approximately 7% of the *N. crassa* genome, including about 1000 fully-covered genes, all of which are transcriptionally quiescent (*Jamieson et al., 2013*; *Galazka et al., 2016*). The greater than 200 H3K27me2/3 domains, which range from 0.5 to 107 kb, are widely distributed throughout the genome but are enriched at subtelomeric regions (*Jamieson et al., 2013*), as also reported for other fungi (*Schotanus et al., 2015*; *Dumesic et al., 2015*; *Studt et al., 2016*; *Connolly et al., 2013*).

In *D. melanogaster*, DNA regulatory regions known as Polycomb Response Elements (PREs) recruit PcG proteins to specific chromatin targets to maintain transcriptional silencing (*Steffen and Ringrose, 2014*). Recently, similar *cis*-acting elements have been detected in *Arabidopsis thaliana*

(*Xiao et al., 2017*). In vertebrates and other organisms, however, the mechanism by which PcG proteins are directed to particular loci is elusive (*Bauer et al., 2016*). We show that the control of H3K27 methylation is fundamentally different from that of epigenetic marks in constitutive heterochromatin. Not only is the genomic distribution of H3K27 methylation much more plastic than that of H3K9 methylation (*Jamieson et al., 2016*; *Mathieu et al., 2005*; *Deleris et al., 2012*; *Lindroth et al., 2008*; *Reddington et al., 2013*; *Hagarman et al., 2013*; *Wu et al., 2010*), but also, unlike DNA methylation and methylation of histone H3K9, which are faithfully methylated *de novo* when introduced at arbitrary ectopic genomic sites (*Miao et al., 1994*; *Selker et al., 1987*), we show that H3K27 methylation is often position-dependent. We further demonstrate that telomere repeats underpin the observed position effect on H3K27me by showing that loss of telomerase abolishes subtelomeric H3K27me2/3 and that artificial introduction of telomere repeats at interstitial sites trigger deposition of H3K27me2/3. That is, telomere repeats themselves are both necessary for subtelomeric H3K27me2/3 and sufficient to trigger ectopic H3K27 methylation at internal chromosomal sites.

## Results

### Analyses of classical chromosome rearrangements suggest that proximity to a chromosome end is key to subtelomeric H3K27 methylation

To search for *cis*-acting sequences that trigger facultative heterochromatin formation in *N. crassa*, perhaps analogous to PREs in *D. melanogaster*, we dissected a 47 kb H3K27me2/3 domain on linkage group (LG) VIL. A series of eight, partially overlapping, three kb fragments from this domain were separately targeted to both the *histidine 3* (*his-3*) and the *cyclosporin A resistance 1* (*csr-1*) euchromatic loci in a strain in which we had deleted the endogenous H3K27me2/3 domain (*Figure 1—figure supplement 1A*). H3K27me2/3 chromatin immunoprecipitation (ChIP) followed by qPCR demonstrated the absence of *de novo* H3K27me2/3 within all eight segments when transplanted to either *his-3* or *csr-1* (*Figure 1—figure supplement 1B and C*).

To address the possibility that the failure to induce H3K27me2/3 in the transplantation experiments was simply due to the size of the test fragments, we utilized strains with large chromosomal rearrangements (*Perkins, 1997*). We first examined translocations that involved the H3K27me2/3 domain on LG VIL that we had dissected, starting with UK3-41, an insertional translocation strain that has an approximately 1.88 Mb segment of LG VR inserted into a distal position on LG VIL (*Figure 1A*). The translocation shifted most of LG VIL to a more interior chromosomal position. Interestingly, H3K27me2/3 ChIP-seq of UK3-41 showed an absence of H3K27me2/3 in the region that was displaced from the chromosome end (lost H3K27me2/3; indicated in orange in *Figure 1A*). This result is consistent with our finding that transplantation of segments from this region did not induce methylation at *his-3* or *csr-1* and suggests that the normal methylation of this subtelomeric domain is position-dependent.

These results raised the question of whether H3K27me2/3 in this domain absolutely depends on its normal location, or whether the methylation would occur if the region were moved near another chromosome end. To address this possibility, we utilized OY350, a quasiterminal translocation that transfers a distal segment of LG VIL to the end of LG IR (*Figure 1B*). Analysis of OY350 by H3K27me2/3 ChIP-seq showed that the translocated LG VIL domain preserved its normal H3K27me2/3 distribution when placed adjacent to the adoptive telomere. This finding supported the hypothesis that H3K27me2/3 in this domain depends on its proximity to the chromosome end and suggested that this requirement is non-specific; the different chromosome end apparently substituted for the native one. A similar situation was observed on LG VIR in OY320 (*Figure 1F*).

The results described above suggested that proximity to a chromosome end may be sufficient to induce H3K27me. Interestingly, in addition to consistent losses of previously subtelomeric H3K27me2/3 in the examined genomic rearrangements, we identified new H3K27me2/3 domains at novel subtelomeric regions. Striking examples of *de novo* H3K27me2/3 were independently observed in seven genome rearrangements at their new chromosome ends (*Figure 1B–E* and *Figure 1—figure supplement 2A–C*; marked in green). These novel domains of H3K27me2/3 extended an average of approximately 180 kb into the new subtelomeric regions. The only apparent exception

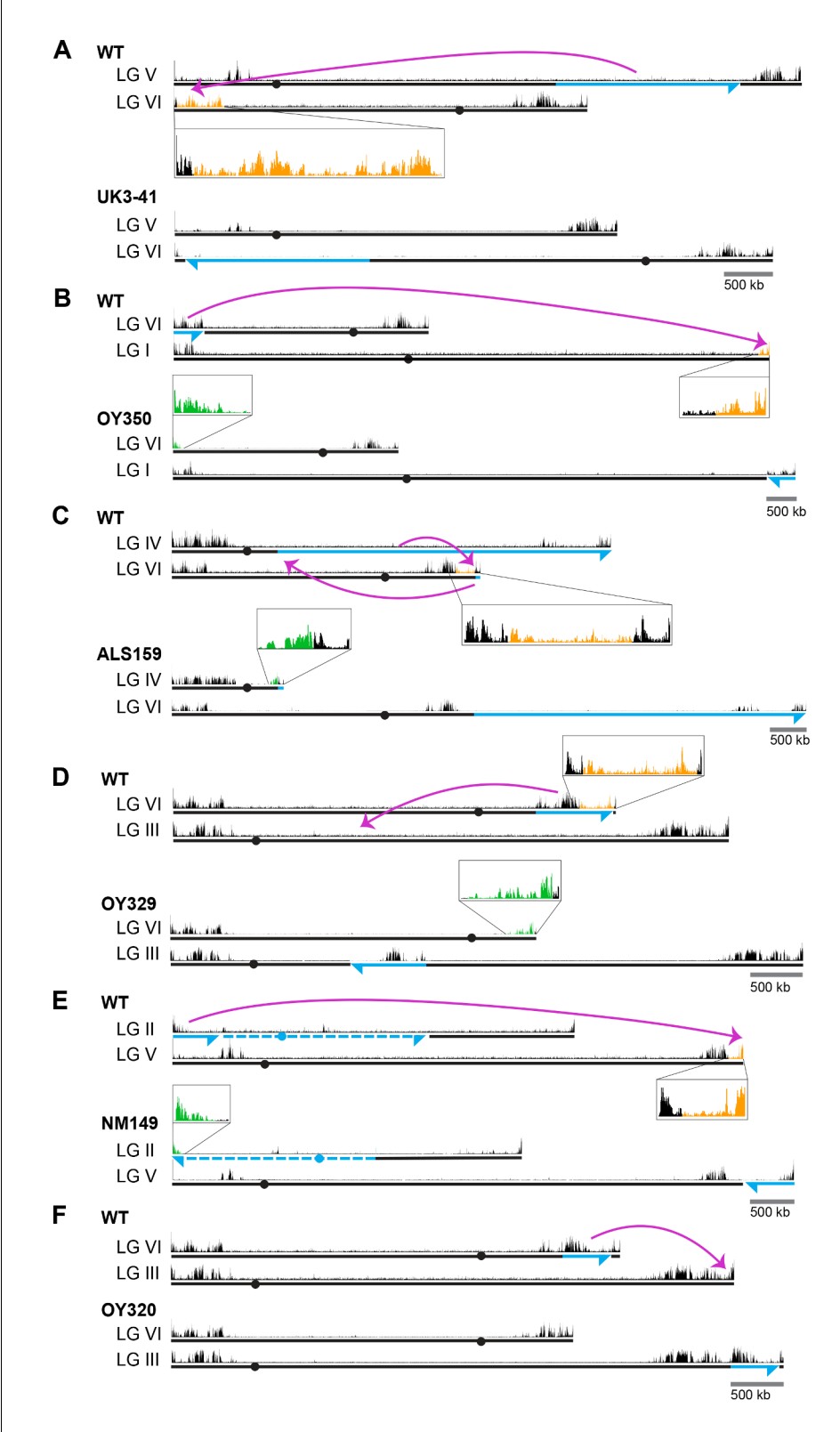

**Figure 1.** Chromosomal rearrangements are associated with altered H3K27me2/3. (A–F) Schematics show the movement (magenta curved arrows) of translocated segments and the resulting chromosomal rearrangements for six translocation strains. ChIPs were done on biological triplicates and pooled for sequencing. Solid and dashed blue lines indicate source of translocated segments while arrowheads indicate directionality of rearranged segments. H3K27me2/3 ChIP-seq tracks of WT and translocation strains are displayed above chromosome diagrams with zoom-in sections in boxes.
*Figure 1 continued on next page*

*Figure 1 continued*

H3K27me2/3 signals that were lost in translocation strains are shown in orange, H3K27me2/3 signals gained are indicated in green and invariant H3K27me2/3 signals are shown in black. Circles indicate centromeres. Gains and losses of H3K27me2/3 were confirmed by qPCR (*Figure 1—figure supplement 4*).

DOI: https://doi.org/10.7554/eLife.31216.002

The following figure supplements are available for figure 1:

**Figure supplement 1.** Three kilobase segments from a natural H3K27me2/3 domain are insufficient to trigger *de novo* H3K27me2/3 at ectopic loci.

DOI: https://doi.org/10.7554/eLife.31216.003

**Figure supplement 2.** H3K27me2/3 profiles of additional strains containing chromosomal rearrangements.

DOI: https://doi.org/10.7554/eLife.31216.004

**Figure supplement 3.** Whole genome view of H3K27me2/3 ChIP-seq in WT and chromosomal translocation strains.

DOI: https://doi.org/10.7554/eLife.31216.005

**Figure supplement 4.** qPCR validation of H3K27me2/3 ChIP-seq data from translocation strains.

DOI: https://doi.org/10.7554/eLife.31216.006

concerns translocation UK2-32 (*Figure 1—figure supplement 2A*), which involved the left end of LG V with approximately one Mb of tandemly repeated rDNA that is not shown in the *N. crassa* genome assembly. The H3K27me2/3 distribution on all chromosomes unrelated to the genome rearrangements was unaltered (*Figure 1—figure supplement 3*). Altogether, our survey of chromosomal translocations strongly suggests that chromosome ends promote the deposition of H3K27me on neighboring chromatin.

## Identification of position-independent H3K27-methylated domains

While domains of H3K27me2/3 are enriched near the ends of chromosomes in *N. crassa*, substantial domains are also found elsewhere. To examine the possibility of position-independent H3K27me2/3 residing within LG VIR, we examined the distribution of H3K27me2/3 in two translocations that effectively moved LG VIR to the middle of a chromosome. ALS159 is a reciprocal translocation that shifted wild-type LG VIR approximately 4.5 Mb away from the new telomere (*Figure 1C*). Although the shifted segment lost the H3K27me2/3 that was previously closest to the chromosome end, the more internal H3K27me2/3 was unaffected by the translocation (*Figure 1C*). Similarly, when LG VIR was inserted in the middle of LG III in OY329 (*Figure 1D*), most of the H3K27me2/3 on LG VIR was retained. Again, the H3K27me2/3 that was originally closest to the telomere was lost, however, consistent with the idea that subtelomeric H3K27me2/3 domains depend on their proximity to the chromosome ends.

To determine if the retention of H3K27me2/3 from LG VIR was unique, we examined the position-dependence of two other H3K27me2/3 domains. In translocation NM149, approximately 600 kb of LG IIL was translocated onto the end of LG VR, effectively shifting the native right arm of LG V about 600 kb away from the chromosome end (*Figure 1E*). Like the behavior of LG VIR in ALS159 and OY329, losses of H3K27me2/3 occurred in the most distal section of LG VR, but most H3K27me2/3 was retained, even when moved away from the chromosome end. In addition, a large H3K27me2/3 domain on LG IIIR was shifted approximately 450 kb away from the chromosome end in OY320, yet it did not incur losses (*Figure 1F*). Taken together, these findings demonstrate that some H3K27me2/3 domains can be maintained when translocated to internal chromosomal sites, *i.e.* some H3K27me domains appear to be position-independent.

## *Cis-trans* test on H3K27 methylation in a segmental duplication strain reveals position effect

The observation that some chromosomal segments that are normally marked with H3K27me2/3 lose the mark when translocated to another genomic location suggested that H3K27me2/3 can be position-dependent. It remained formally possible, however, that the loss of H3K27me2/3 was an indirect effect of the translocation, perhaps due to an altered transcriptional program. To examine the possibility that one or more *trans*-acting factors could be responsible for the loss of H3K27me2/3 in OY329, we crossed this strain, which is in an Oak Ridge (OR) background, to a highly polymorphic wild-type strain, Mauriceville (MV), to obtain progeny with a segmental duplication, that is, with the OR translocated LG VIR domain inserted into LG II in a strain that also contained the corresponding

region on the native LG VI of MV (*Figure 2A*). The high density of single nucleotide polymorphisms (SNPs) in the MV background allowed us to separately map the H3K27me2/3 distribution in the translocated (OR) segment and the wild-type (MV) segment (*Pomraning et al., 2011*). SNP-parsed H3K27me2/3 ChIP-seq showed that the duplicated chromosomal segments have distinct H3K27me2/3 profiles (*Figure 2B*). The previously subtelomeric H3K27me2/3 that was lost in OY329 did not return in the duplication strain, nor was the loss in this region recapitulated on the native LG VIR of MV. The independent behavior of these homologous segments suggests that the loss of H3K27me2/3 in OY329 was not a result of *trans*-acting factors, but rather was a result of the translocation itself, that is, it is a *bona fide* position effect.

## Altered gene expression in regions with changed H3K27me2/3

Considering that H3K27me2/3 normally marks transcriptionally quiescent chromatin in *N. crassa* (*Jamieson et al., 2013*), it was of obvious interest to ascertain whether ectopic H3K27me2/3 resulting from chromosomal translocations could cause gene silencing. To determine if the loss or gain of H3K27me2/3 in the translocation strains activated or repressed gene expression, respectively, we performed poly-A+ mRNA sequencing on wild-type and seven translocation strains (ALS159, AR16, OY329, OY337, OY350, UK2-32, and UK3-41). *Figure 3A and B* show gene expression and associated H3K27me2/3 levels in the wild-type and translocation OY350 strains at representative regions where former subtelomeres were moved and new ones created. The gain of H3K27me2/3 on LG VIL in OY350, due to formation of a novel subtelomere, coincided with reduced transcript levels relative to wild-type (*Figure 3A*). Conversely, the loss of H3K27me2/3 on LG IR in OY350, due to shifting the

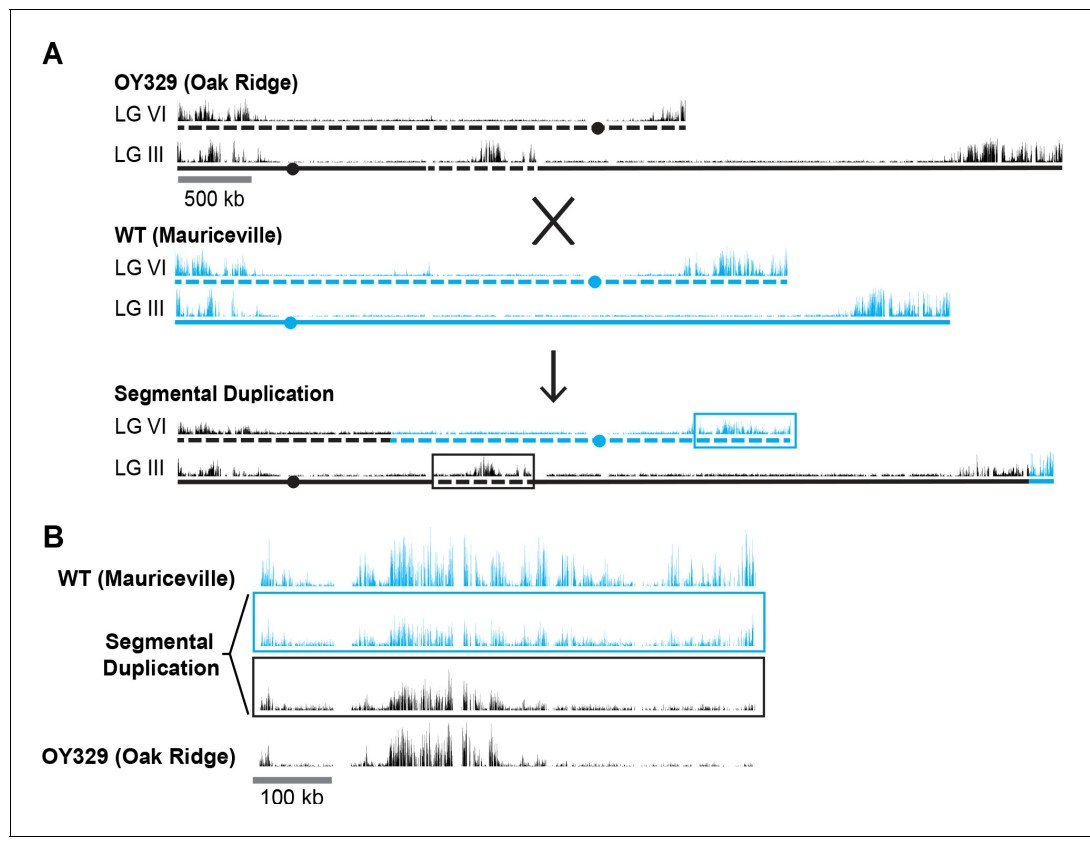

**Figure 2.** A segmental duplication confirms H3K27me2/3 position effect. (**A**) Diagram of a cross between the OY329 (Oak Ridge) insertional translocation strain and a polymorphic wild-type strain (Mauriceville), resulting in a strain bearing a duplicated chromosomal segment (outlined by rectangles). H3K27me2/3 ChIP-seq tracks (single experiment) and chromosomes are shown in black for the Oak Ridge strain and blue for Mauriceville. Solid and dashed lines indicate chromosome source. (**B**) Expanded view of the duplicated chromosome segment in parental strains and duplication-containing offspring along with associated SNP-parsed H3K27me2/3 profiles.

DOI: https://doi.org/10.7554/eLife.31216.007

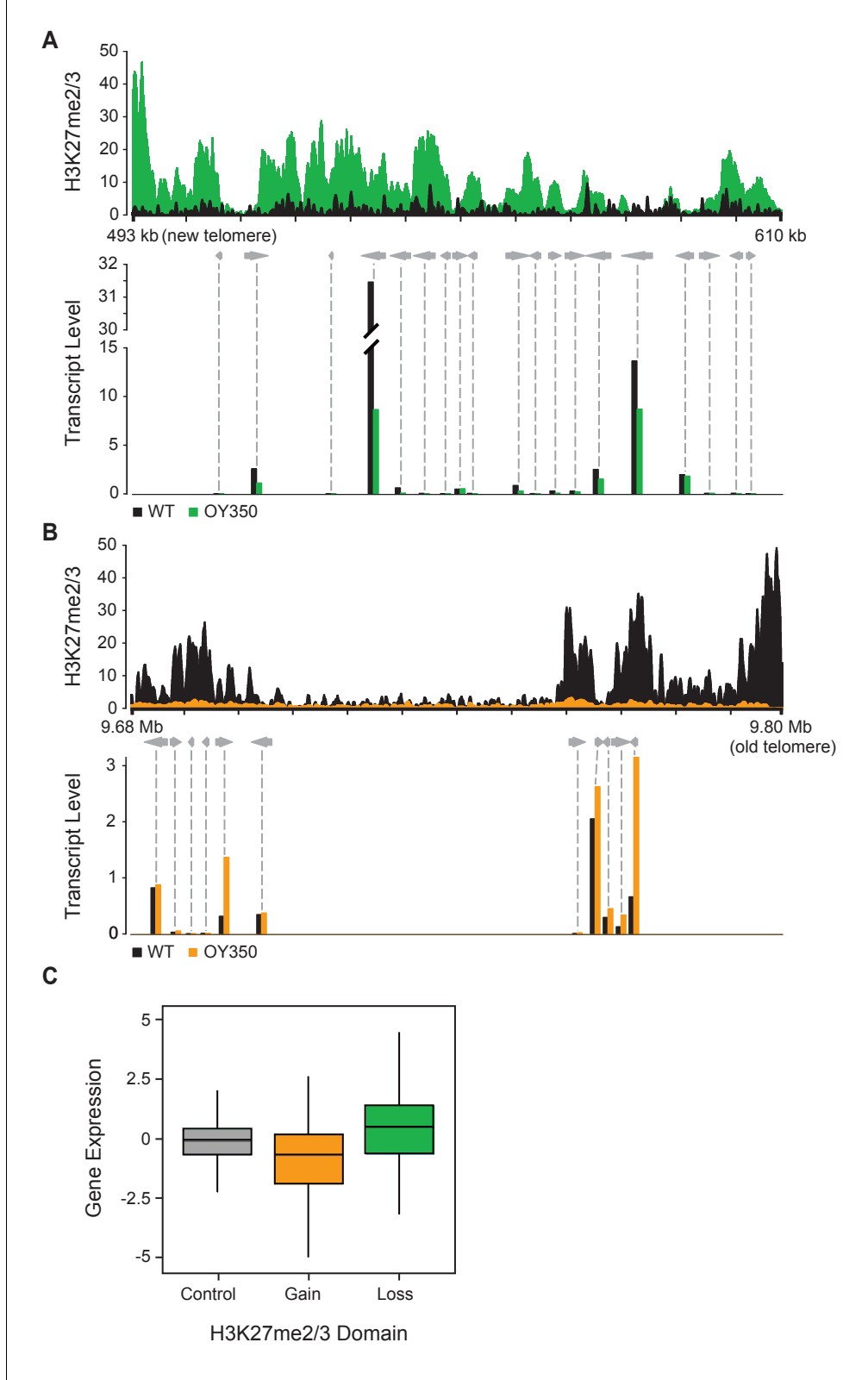

**Figure 3.** Altered gene expression reflects changes in H3K27me2/3. (**A**) Reduced gene expression in novel subtelomeric region (LG VIL) that gains H3K27me2/3 in translocation OY350. The top panel displays H3K27me2/3 ChIP-seq reads of OY350 (green) normalized to the corresponding non-translocated segment in WT (black) using a 350 bp sliding window. The bottom panel shows normalized mRNA-seq read counts (counts/1000) for genes (gray arrows) in the region for WT (black) and OY350 (green). ChIP-seq data are from pooled biological triplicates; RNA-seq data are from

*Figure 3 continued on next page*

*Figure 3 continued*

biological duplicates. (B) Shifting a normally subtelomeric region to an internal position in translocation OY350 results in loss of H3K27me2/3 on LG IR and increased gene expression. The top panel displays H3K27me2/3 ChIP-seq reads of OY350 (orange) normalized to WT (black) using a 500 bp sliding window. The bottom panel shows normalized mRNA-seq read counts (counts/1000) for genes (gray arrows) in the region for WT (black) and OY350 (orange). (C) A box plot summarizes the relationship between gain (green) or loss (orange) of H3K27me2/3 in seven chromosomal translocation strains (ALS159, AR16, OY329, OY337, OY350, UK2-32, and UK3-41) and associated gene expression changes (log$_2$[translocation/WT]) within the corresponding domains. The control (gray) represents gene expression in regions that do not exhibit changes in H3K27me2/3.

DOI: https://doi.org/10.7554/eLife.31216.008

The following figure supplement is available for figure 3:

**Figure supplement 1.** qPCR validation of RNA-seq expression changes in translocation OY350.

DOI: https://doi.org/10.7554/eLife.31216.009

subtelomeric region, was associated with an increase in transcript abundance compared to wild-type (*Figure 3B*). Results of qPCR analyses of select genes confirmed these findings (*Figure 3—figure supplement 1*). Indeed, a comprehensive analysis of poly-A+ mRNA-seq data from all seven translocation strains showed increases in gene expression at previously subtelomeric regions that lost H3K27me2/3 and decreases in gene expression at novel subtelomeres that gained H3K27me2/3 (*Figure 3C*). These findings demonstrate that chromosomal rearrangements can cause marked changes in both H3K27me2/3 and gene expression of the affected regions.

## Loss of telomerase abolishes subtelomeric H3K27me2/3

Our analyses of translocation strains demonstrated that domains of H3K27me2/3 can be position-dependent and suggested that a feature of chromosome ends directly or indirectly recruits H3K27me2/3 to subtelomeric regions. To investigate the basis of this, we utilized a mutant lacking the single telomerase reverse transcriptase (*tert*) that is responsible for all (TTAGGG)$_n$ telomere repeats normally found on *N. crassa* chromosome ends (*Wu et al., 2009*). Southern analysis of the *tert* strain using a (TTAGGG)$_n$ probe revealed major reductions in the hybridization signals and showed that the majority of telomere fragments present in wild-type were either undetectable or greatly reduced in the mutant (*Figure 4A*). In *S. pombe*, the majority of *trt1*$^-$ (yeast homologue of *N. crassa tert*) survivors have circularized their chromosomes (*Nakamura et al., 1998*). To determine if *N. crassa tert* survivors also have circular chromosomes, we designed outwardly directed PCR primers to amplify sequences near the ends of each chromosome and tested whether fragments were generated by fusions of the right and left chromosome ends. Indeed, the *tert* genomic DNA, but not control DNA from a wild-type strain, supported amplification using the divergent primer pairs, confirming intra-chromosomal fusions (*Figure 4B*).

We performed H3K27me2/3 ChIP-seq in biological duplicate to determine whether the subtelomeric domains of H3K27me2/3 were retained or lost in the *tert* strain and found clear evidence of loss of H3K27 methylation from chromosome ends (*Figure 4C*). With the exception of LG V, which is unique in that one of its ends is capped with approximately 150 copies of the approximately 9 kb rDNA repeat (*Butler and Metzenberg, 1990*), all chromosomes lost their subtelomeric H3K27me2/3 domains, which typically extend tens of thousands of bp from their ends. Internal H3K27me2/3 domains were not noticeably altered (*Figure 4C* and *Figure 4—figure supplement 1*). Sequencing of *tert* input DNA from the ChIP showed no reduction in coverage near the chromosome ends, confirming that the losses of H3K27me2/3 ChIP signal observed in *tert* are not caused by chromosome degradation. We conclude that either (TTAGGG)$_n$ telomere repeats or linear chromosome ends are required for position-dependent domains of H3K27me2/3 in *N. crassa*.

## Internal telomere repeats are sufficient to induce *de novo* H3K27me2/3

To determine if the presence of telomere repeats is sufficient to trigger the deposition of H3K27 methylation, telomere repeats were separately targeted to the *csr-1* and *his-3* loci. Due to the repetitive nature of these sequences and our cloning strategy, strains containing a variable number of (TTAGGG) repeats could be obtained through homologous recombination. Sequencing of inserted DNA in eleven independent transformants revealed that the number of inserted telomere repeats ranged from 5 to 23. Native *N. crassa* telomeres have an average of 20 repeats (*Wu et al., 2009*). H3K27me2/3 ChIP-qPCR demonstrated that ectopic telomere repeats are sufficient to induce local

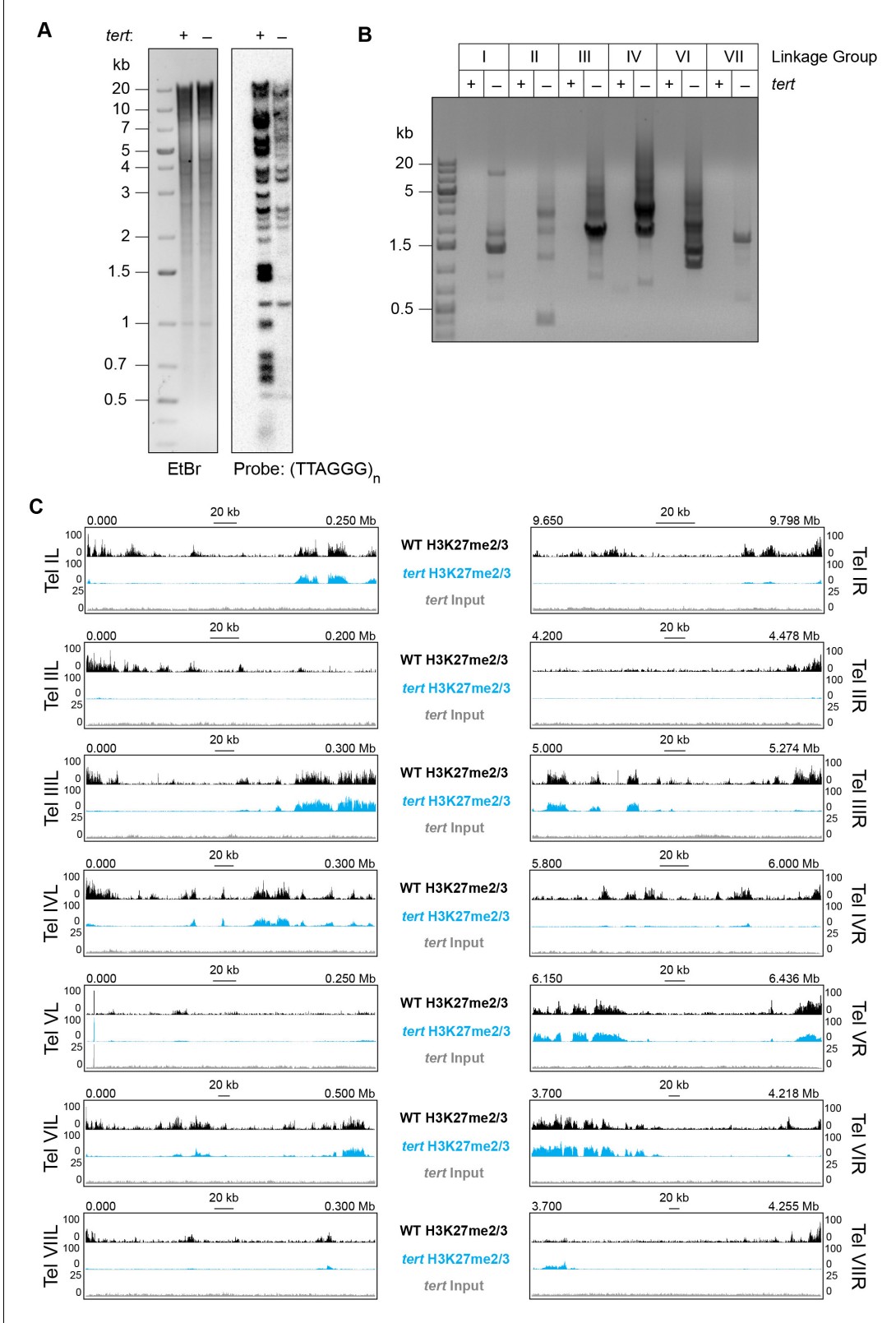

**Figure 4.** Loss of *tert* disrupts chromosome ends and abolishes subtelomeric H3K27me2/3. (**A**) Southern hybridization of genomic DNA from WT and *tert* strains digested with HindIII/NotI reveals loss of chromosome ends marked with the telomere repeats (TTAGGG)n, which was used as the probe. The ethidium bromide (EtBr) image demonstrates equal loading of WT and *tert* genomic DNA. (**B**) Circularization of chromosomes was demonstrated by generation of PCR products with outwardly directed primers near chromosome ends in a *tert* strain, but not a WT strain. (**C**) H3K27me2/3 ChIP-seq

*Figure 4 continued on next page*

*Figure 4 continued*

of *tert* (single sample; blue track) compared to WT (pooled biological triplicates; black track). Sequence coverage of the *tert* ChIP input (gray track) is also shown.

DOI: https://doi.org/10.7554/eLife.31216.010

The following figure supplement is available for figure 4:

**Figure supplement 1.** Whole genome view of H3K27me2/3 ChIP-seq in WT and *tert*.

DOI: https://doi.org/10.7554/eLife.31216.011

H3K27me2/3 at the *csr-1* and *his-3* loci (*Figure 5A* and *Figure 5—figure supplement 1*). Even insertion of $(TTAGGG)_8$ triggered some methylation and insertion of $(TTAGGG)_{17}$ led to a high level of H3K27me2/3 at *csr-1*. ChIP-seq on the $(TTAGGG)_{17}$ strain revealed new peaks as far as 170 kb from the insertion site and the semi-continuous H3K27me2/3 domain spanned approximately 225 kb including 30 genes (*Figure 5B* and *Figure 5—figure supplement 2*). This is comparable to the size of subtelomeric H3K27me2/3 domains that were lost in the *tert* strain.

## Discussion

The proper distribution of the facultative heterochromatin mark, H3K27me2/3, deposited by PRC2, is necessary for appropriate gene expression in a variety of plants, animals and fungi (*Wiles and Selker, 2017*). Unfortunately, the control of H3K27 methylation remains largely unknown. In *D. melanogaster*, DNA elements known as PREs are important to define domains of H3K27me and associated silencing, but even in this organism, PREs are neither sufficient in all genomic contexts, nor fully penetrant (*Cunningham et al., 2010*; *Horard et al., 2000*). In addition, it was recently found that deletion of a well-studied PRE in *D. melanogaster* did not significantly affect either gene silencing or H3K27 methylation at its native locus (*De et al., 2016*). The control of H3K27 methylation is less defined in mammals, in which only a few candidate PRE-like elements have been identified (*Basu et al., 2014*; *Sing et al., 2009*; *Woo et al., 2010*).

We took advantage of a relatively simple eukaryote bearing H3K27me, the filamentous fungus *N. crassa*, to explore the control of this chromatin mark. Conveniently, H3K27me is non-essential in this organism, in spite of being responsible for silencing scores of genes (*Klocko et al., 2016*; *Jamieson et al., 2013*). Unlike the situation with constitutive heterochromatin in *N. crassa* (*Miao et al., 1994*; *Selker et al., 1987*), we demonstrated that not all facultative heterochromatin is entirely controlled by underlying sequence elements; transplanted gene-sized segments of a H3K27me domain do not, in general, become faithfully H3K27-methylated. However, translocations of large chromosomal segments revealed two distinct classes of transcriptionally repressive H3K27me domains, position-dependent and position-independent. Mechanistic insights from subsequent experiments defined these classes as telomere repeat-dependent and telomere repeat-independent H3K27me. The identification of telomere repeats as potent signals for H3K27 methylation represents a major advance in our understanding of facultative heterochromatin formation in eukaryotes.

Our study took advantage of a collection of spontaneous and UV-induced chromosome rearrangement strains of *N. crassa* that were primarily collected and characterized by David Perkins (*Perkins, 1997*). We surveyed representative rearrangement strains in which chromosomal regions containing domains of H3K27me were translocated to novel genomic positions. Strikingly, ChIP-seq revealed multiple cases in which subtelomeric H3K27me2/3 domains completely lost this modification when moved to an interior chromosomal location. Indeed, no exceptions to this rule were found. Generation and analysis of a segmental duplication strain containing differentially marked translocated and normal segments confirmed that the changed distribution of H3K27me2/3 was a *bona fide* position effect rather than an effect of a *trans*-acting factor (*Figure 2*). Moreover, new subtelomeric regions gained H3K27me2/3 over sequences that were previously devoid of this mark (*Figure 1*). We conclude that proximity to a chromosome end, per se, somehow induces deposition of H3K27me2/3 in domains that can span hundreds of kilobases. Analyses of polyA+ mRNA from the strains revealed reductions in gene expression associated with the new H3K27me2/3 and increases in gene expression in regions that lost this mark; that is, the changes in H3K27me2/3 were reflected in underlying gene expression levels (*Figure 3*).

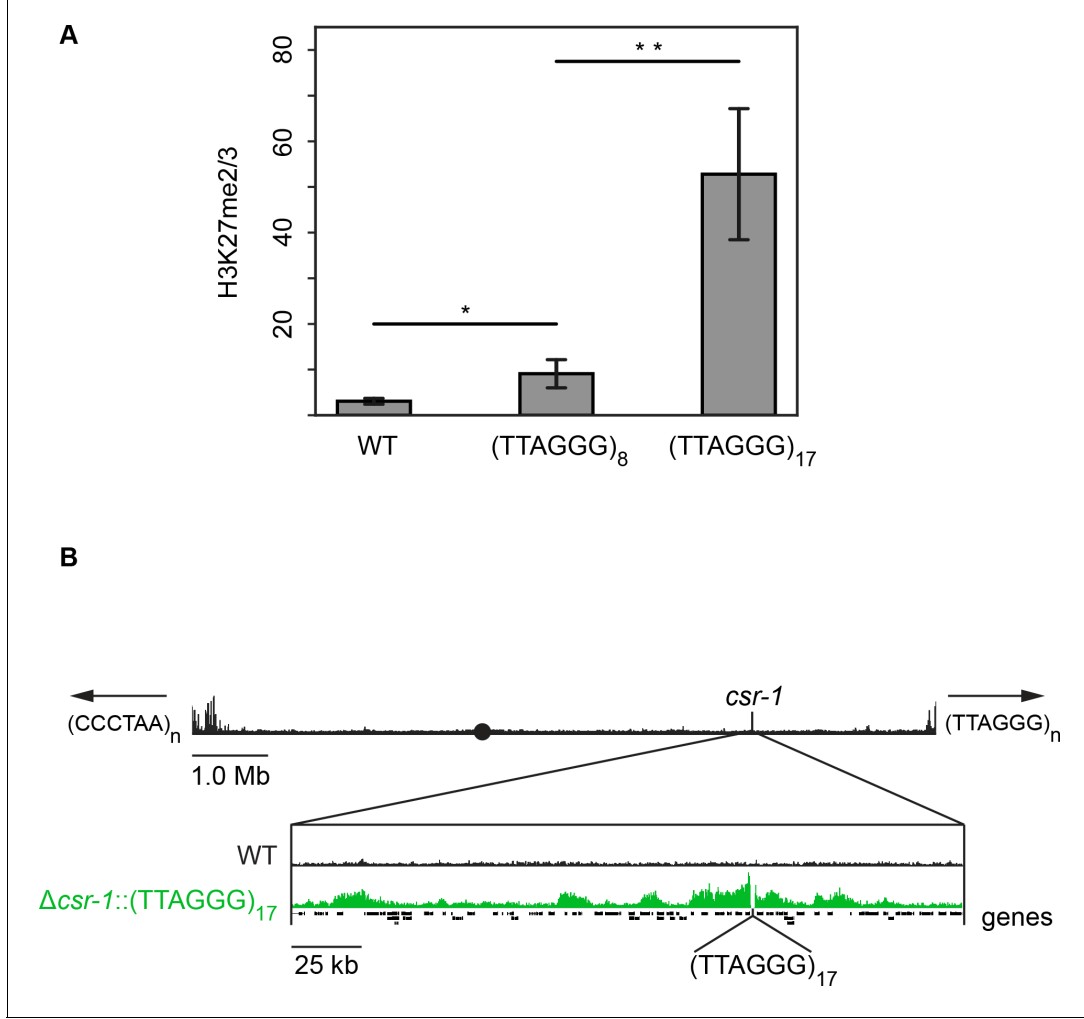

**Figure 5.** Telomere repeats targeted to the *csr-1* locus induce an approximately 225 kb H3K27me2/3 domain. (A) qPCR analyses (biological triplicates) of H3K27me2/3 ChIP with strains containing 8 or 17 telomere repeats at the *csr-1* locus, with WT strain as a control (means displayed; error bars show standard deviation; single and double asterisks represent p=0.01488 and p=0.00336 respectively). Level of H3K27me2/3 normalized to Telomere 1L. $M_{WT}$ = 3.05970, $SD_{WT}$ = 0.64784, 95% $CI_{WT}$ = [2.32656, 3.79264]; $M_8$ = 9.08627, $SD_8$ = 3.09022, 95% $CI_8$ = [5.58937, 12.58303]; $M_{17}$ = 52.78110, $SD_{17}$ = 14.35716, 95% $CI_{17}$ = [36.53461, 69.02719]. (B) H3K27me2/3 ChIP-seq of full WT LG I (pooled biological triplicates; black track). Black circle indicates the centromere, *csr-1* position is indicated by vertical line, and arrows at chromosome ends indicate 5' to 3' polarity of telomere repeats. Expansion below shows the extent and shape of the H3K27me2/3 domain induced in the Δ*csr-1*::(TTAGGG)$_{17}$ strain (single sample; green track) compared to WT (black track). Site of *csr-1* replacement with telomere repeats indicated below ChIP-seq tracks. Genes are displayed as black bars.
DOI: https://doi.org/10.7554/eLife.31216.012

The following figure supplements are available for figure 5:

**Figure supplement 1.** Telomere repeats targeted to the *his-3* locus induce local H3K27me2/3 domain.
DOI: https://doi.org/10.7554/eLife.31216.013

**Figure supplement 2.** Whole genome view of H3K27me2/3 ChIP-seq in WT and Δ*csr-1*::(TTAGGG)$_{17}$.
DOI: https://doi.org/10.7554/eLife.31216.014

Although gene silencing near chromosome ends, sometimes dubbed 'telomere position effect' (TPE) has been observed in fungi (*Kyrion et al., 1993*; *Nimmo et al., 1994*; *Castaño et al., 2005*; *Smith et al., 2008*; *Shaaban et al., 2010*), *D. melanogaster* (*Doheny et al., 2008*), *Mus musculus* (*Pedram et al., 2006*), and human cells (*Baur et al., 2001*), previous studies did not implicate H3K27me. Interestingly, the extent of heterochromatin spreading from telomeres in *N. crassa* is substantially greater than previously reported. This could be due to the conspicuous absence of

canonical subtelomere sequences in *N. crassa* (*Wu et al., 2009*; *Mefford and Trask, 2002*; *Arnoult et al., 2012*; *Pryde and Louis, 1999*). Not all organisms that exhibit TPE have H3K27 methylation machinery, but in *D. melanogaster*, which does, systematic analysis of PcG mutants demonstrated they do not disrupt telomere silencing (*Doheny et al., 2008*). In human cells, TPE is thought to be mediated by histone deacetylation, H3K9 methylation, and heterochromatin protein HP1α (*Tennen et al., 2011*; *Arnoult et al., 2012*). Although the phenomenon of TPE has been studied for decades, our findings may reflect the first documented case of H3K27me-mediated TPE.

The ability of chromosome ends to induce large domains of H3K27me2/3 in chromosomal translocation strains motivated us to investigate the role of telomere sequences in the establishment of facultative heterochromatin. We found that deletion of *tert*, the sole telomerase reverse transcriptase gene, results in a dramatic loss of H3K27me2/3 at subtelomeres, concomitant with loss of (TTAGGG)$_n$ repeats and chromosome circularization (*Figure 4*). To directly test the possibility that telomere repeats can trigger domains of H3K27me2/3, we inserted an array of telomere repeats at two euchromatic, interstitial sites and used ChIP to check induction of H3K27me2/3. We found that, indeed, even a 48 bp array, (TTAGGG)$_8$, could induce some local H3K27me2/3 at the *csr-1* locus. The independent induction of local H3K27me2/3 at the *his-3* locus suggests this effect is not position-specific (*Figure 5—figure supplement 1*). Remarkably, a 102 bp array, (TTAGGG)$_{17}$, induced an H3K27me2/3 domain that covered approximately 225 kb, including 30 genes (*Figure 5*). These results strongly suggest that wild-type subtelomeric H3K27me2/3 is dependent on telomere repeats. The recruitment of PRC2 to telomere repeats may be a widespread phenomenon, considering that enrichment of H3K27 methylation at telomeres has been observed in fungi (*Jamieson et al., 2013*; *Schotanus et al., 2015*; *Dumesic et al., 2015*; *Studt et al., 2016*), plants (*Baker et al., 2015*; *Vaquero-Sedas et al., 2012*) and animals (*Wirth et al., 2009*). Indirect recruitment of PRC2 by telobox-binding transcription factors (*Xiao et al., 2017*) and direct binding of PRC2 to G-quadruplex RNA resulting from transcription of telomere repeats (*Wang et al., 2017*) represent possible molecular mechanisms. Indeed, it was recently reported that telomeric repeat-containing RNAs (TERRA) directly interact with EZH2 and that H3K27me3 ChIP-seq peaks are strongly correlated with TERRA-associated chromatin in mouse embryonic stem cells (*Chu et al., 2017*).

In addition to telomere repeats capping the ends of chromosomes, telomere repeat-like elements are scattered interstitially in the genomes of many organisms (*Ruiz-Herrera et al., 2008*). In *N. crassa*, interstitial telomere sequences are rare, limited to less than four tandem repeats, and are not preferentially associated with H3K27me2/3 regions. While *S. cerevisiae* and *S. pombe* lack PRC2 components and H3K27me, their internal telomere repeat-like elements can promote heterochromatin formation (*Zofall et al., 2016*; *Duan et al., 2016*). The genome of *Nicotiana tabacum* also has heterochromatic internal telomere repeats, but they lack the H3K27 methylation present at genuine telomeres (*Majerová et al., 2014*). Curiously, insertion of about 130 telomere repeats in Chinese ovary cells failed to significantly alter local transcription (*Kilburn et al., 2001*). It should be interesting to study the effects of terminal and interstitial telomere repeats on heterochromatin formation in a variety of organisms.

We inferred the existence of position-independent H3K27 methylation from our observation that some H3K27me2/3 domains were unaffected when moved to ectopic chromosomal locations (e.g. see domains on LG IIIR in OY320, LG VR in NM149, and LG VIR in ALS159 and OY329; *Figure 1*). The recapitulation of normal H3K27me2/3 profiles at ectopic chromosomal sites is consistent with the possibility that these chromosomal regions contain *cis*-acting signals that trigger the deposition of H3K27 methylation, perhaps comparable to PREs in *D. melanogaster*. It will be of interest to define the presumptive elements responsible for such position-independent H3K27me. While there is no consensus sequence for PREs in *D. melanogaster*, they are composed of unique combinations of binding sites for a variety of factors (*Kassis and Brown, 2013*). One DNA-binding factor known to affect PcG silencing in *D. melanogaster*, GRH, has an obvious homolog in *N. crassa* and could be a candidate for helping establish or maintain position-independent domains of H3K27me2/3 *via* sequence-specific interactions (*Blastyák et al., 2006*; *Nevil et al., 2017*; *Paré et al., 2012*). It remains possible that deposition of H3K27 methylation at position-independent domains is not directed by sequence-specific elements. In this context, it is worth noting that inhibition of transcription, caused by either exposure to RNA polymerase II inhibitors (*Riising et al., 2014*) or deletion of a transcriptional start site (*Hosogane et al., 2016*) was found to promote the deposition of H3K27me in mammalian cells. Conceivably, the position-independent domains of H3K27me2/3 in *N.*

*crassa* could be directed by blocs of inherently low-expressing genes or by some other feature of the region, such as its location in the nucleus, which may be controlled by yet undefined factors. The fact that only about 10% of H3K27me2/3-marked genes are upregulated when the sole H3K27 methyltransferase is removed supports the notion that transcriptional shut-off may precede PRC2 recruitment (*Klocko et al., 2016*; *Jamieson et al., 2013*). Still, low transcriptional activity cannot entirely explain the targeting of PRC2 in *N. crassa* since many low-expressed genes are not H3K27 methylated.

Genome rearrangements are a common occurrence in malignant cells and can drive tumorigenesis through the creation of gene fusions, enhancer hijacking and oncogene amplification (*Hnisz et al., 2016*). Our findings suggest that genome rearrangements associated with cancer may additionally impact gene expression through effects on facultative heterochromatin. Indeed, there are already some reports of human diseases being driven by position effects (*Surace et al., 2014*; *Guilherme et al., 2016*). A great deal of research has focused on how characteristics of local chromatin influence translocation breakpoint frequencies (*Hogenbirk et al., 2016*), but the effects of the resulting translocations on chromatin state are still ill-defined. Altogether, our work shows that changes in genome organization can have sweeping effects on both the distribution of an epigenetic mark and gene expression. Thus, chromosome rearrangements may have unappreciated roles in evolution and cancer etiology.

## Materials and methods

### Strains, media and growth conditions

*N. crassa* strains are listed in *Supplementary file 1* and were grown and maintained according to standard procedures (*Davis, 2000*). All genome rearrangement strains are available through the Fungal Genetics Stock Center (www.fgsc.net).

### Deleting and targeting segments of LG VIL

A strain containing a 47.4 kb deletion of LG VIL (N4933) was constructed by homologous recombination using primers listed in *Supplementary file 2* (*Colot et al., 2006*). For *his-3* constructs, segments from LG VIL were PCR-amplified from wild-type (N3752) genomic DNA with primers containing restriction enzyme sites (*Supplementary file 2*) and directly cloned into pCR (Life Technologies TA Cloning Kit), subcloned into the *his-3*-targeting vector pBM61 (*Supplementary file 3*) and transformed into *N. crassa* strain N5739 (*Margolin et al., 1997*). For *csr-1* replacements, segments from LG VIL were PCR-amplified from wild-type (N3752) genomic DNA with primers containing homology to the 5' and 3' flanks of the *csr-1* locus (*Supplementary file 2*). Amplified segments were subsequently assembled by PCR ('PCR-stitched') to the 5' and 3' flanks of the *csr-1* locus respectively. Stitched PCR products were co-transformed into *N. crassa* strain N2931 and transformants were selected on cyclosporin A-containing medium (*Bardiya and Shiu, 2007*).

### Knocking-out *tert*

The *nat1* gene with the *trpC* promoter was amplified by PCR using the pAL12-Lifeact as the template with primers 3902 and 1369. The 5' and 3' flanking fragments of the *tert* gene were amplified by PCR with primers 3406–3409, which have specific 29 bp overhang sequence with the 5' and 3' *nat1* gene with the *trpC* promoter. The three PCR products were gel-purified, combined and PCR-stitched with primers 3406 and 3409 to construct the knockout cassette. The cassette was gel-purified and transformed into a Δ*mus-51* strain (N2929) by electroporation and resulting strains were crossed with a *mus-51*⁺ strain (N3752) to recover progeny with the wild-type allele of *mus-51*.

### Targeting telomere repeats to *csr-1* and *his-3*

Concatamers of telomere repeats were amplified in a polymerase chain reaction lacking template as previously described (*Wu et al., 2009*). Resulting PCR products of ~500 bp were gel-purified and cloned into the pCR4-TOPO TA vector (Life Technologies TOPO TA Cloning kit for Sequencing). Telomere repeats from the cloning vector were PCR-stitched separately to 5' and 3' flanks of the targeting locus. For *csr-1* targeting, stitched PCR products were co-transformed into strain N5739 and homokaryotic transformants were selected on cyclosporin A-containing medium (*Bardiya and Shiu,*

*2007*). For *his-3* targeting, stitched PCR products were co-transformed into strain N2834 and heterokaryotic transformants were selected on medium lacking histidine.

## ChIP and preparation of libraries

ChIP was performed as previously described (*Jamieson et al., 2016*). An anti-H3K27me2/3 antibody (Active Motif, 39535) was used for all experiments. ChIPs for *Figures 1* and *3*, and *Figure 1—figure supplements 1–3* were performed in biological triplicate. Real-time qPCR was performed as previously described (*Jamieson et al., 2013*). ChIP-seq libraries were also prepared as previously described (*Jamieson et al., 2016*), except N51, N6089, N6228 and N6383 libraries were subjected to eight cycles of amplification. Sequencing was performed using the Illumina NextSeq 500 using paired-end 100 nucleotide reads for input chromatin and some ChIP experiments. The remainder of the ChIP experiments were sequenced using the Illumina HiSeq 2000 using single-end 100 nucleotide reads or the NextSeq500 using single-end 75 nucleotide reads. All sequences were mapped to the corrected *N. crassa* OR74A (NC12) genome (*Galazka et al., 2016*) using Bowtie2 (*Langmead and Salzberg, 2012*). ChIP-seq read coverage was averaged over 100 bp sliding (50 bp increment) windows with BEDTools (*Quinlan and Hall, 2010*) and normalized to wild-type by coverage. ChIP-seq data were visualized with Gviz (*Hahne and Ivanek, 2016*). To adjust for different sequencing depths between wild-type and translocation samples, a scaling factor was determined for each sample as a ratio of total number of mapped reads in the given sample relative to wild-type. Each scaling factor was then used to normalize ChIP-seq samples. ChIP-seq data were visualized with either IGV (*Thorvaldsdóttir et al., 2013*) for *Figures 1–2* and *4–5*, *Figure 1—figure supplements 1A*, *2* and *3*, *Figure 4—figure supplement 1* and *Figure 5—figure supplement 2* or Gviz for *Figure 3A and B* (*Hahne and Ivanek, 2016*). Normalized H3K27me2/3 ChIP-seq data were displayed using a 350 or 500 bp sliding widow unless otherwise specified in the figure legend.

## RNA isolation and preparation of libraries

RNA isolation and poly(A)-RNA enrichment, RNA-seq library preparation, and subsequent differential expression analysis were performed with replicate samples as previously described (*Klocko et al., 2016*). RNA-seq data were visualized with Gviz (*Hahne and Ivanek, 2016*).

## Translocation breakpoint analyses

Incongruous-paired and split-read alignments for input DNA samples were found using bwa-mem. The discordant and split-read alignments were analyzed with LUMPY v0.2.9 to determine the location of chromosomal breakpoints (*Layer et al., 2014*). A list of predicted breakpoints with a minimum call weight of five was generated and calls with an evidence set score <0.05. The chromosomal breakpoints were compared to mapping data determined by recombination mapping (*Perkins, 1997*), RFLP coverage and inverse PCR (http://hdl.handle.net/10603/68) to remove calls that were not associated with the translocation of interest. PCR analyses of genome rearrangements ALS159, NM149, OY329, OY320 and UK3-41 were consistent with predicted breakpoint patterns (*Supplementary file 4*).

## Southern analysis

Southern hybridization analyses were performed as previously described (*Miao et al., 2000*). *N. crassa* telomere restriction fragments were examined as described previously with a HindIII/NotI double digest (*Wu et al., 2009*). Primers used to make probe are listed in *Supplementary file 2*.

## SNP mapping

ChIP-seq data for the duplication strain were converted into fasta format, trimmed to 70 nt, and filtered for 70 nt long reads using FASTX-Toolkit (http://hannonlab.cshl.edu/fastx_toolkit). Processed data were SNP-parsed with Hashmatch (*Filichkin et al., 2010*) using the Mauriceville-Oakridge SNPome (fasta file listing both versions of each SNP) (*Pomraning et al., 2011*). The SNPome file was modified to align with read lengths of 70 nt. Only reads with 100% alignment to either genome were kept for further analysis and were allocated into two files according to SNP mapping. Each file was then converted back into fastq format using SeqTK 1.0 (https://github.com/lh3/seqtk) and remapped with Bowtie2 to the genome corresponding with its SNP alignment. To adjust for the

variable mapping efficiency of each SNP-parsed file to either Mauriceville or Oakridge, samples were normalized using a scaling factor. The scaling factor was calculated as a ratio of the number of alignments for each SNP-parsed file relative to the file with the smallest number of alignments. H3K27me2/3 profiles were displayed in IGV (*Thorvaldsdóttir et al., 2013*) to examine the individual H3K27me2/3 profiles for wild-type and rearranged chromosome segments.

## Data availability

All ChIP-seq and RNA-seq data, as well as whole genome sequence data, will be available from the NCBI Gene Expression Omnibus (GEO) database (accession GSE104019).

## Acknowledgements

We thank Heejeung Yoo for genotyping *csr-1* targeting strains. This study was supported by a National Institutes of Health grant (GM093061) to EUS and a grant to SH from the Japanese Program to Disseminate Tenure Tracking System, Ministry of Education, Culture, Sports, Science and Technology.

## Additional information

### Funding

| Funder | Grant reference number | Author |
|---|---|---|
| National Institutes of Health | T32 HD007348 | Kevin J McNaught |
| Japanese Program to Disseminate Tenure Tracking System | | Shinji Honda |
| National Institutes of Health | GM093061 | Eric U Selker |

The funders had no role in study design, data collection and interpretation, or the decision to submit the work for publication.

### Author contributions

Kirsty Jamieson, Conceptualization, Formal analysis, Supervision, Funding acquisition, Validation, Investigation, Visualization, Methodology, Writing—original draft, Project administration, Writing—review and editing; Kevin J McNaught, Conceptualization, Data curation, Formal analysis, Validation, Investigation, Visualization, Methodology, Writing—original draft, Writing—review and editing; Tereza Ormsby, Conceptualization, Data curation, Formal analysis, Validation, Investigation, Visualization, Writing—original draft, Writing—review and editing; Neena A Leggett, Data curation, Software, Investigation, Visualization, Writing—review and editing; Shinji Honda, Resources, Data curation, Software, Funding acquisition, Visualization, Writing—review and editing; Eric U Selker, Conceptualization, Resources, Supervision, Funding acquisition, Visualization, Methodology, Writing—original draft, Project administration, Writing—review and editing

### Author ORCIDs

Kirsty Jamieson (ID) https://orcid.org/0000-0003-3110-7081
Kevin J McNaught (ID) http://orcid.org/0000-0002-6887-3161
Eric U Selker (ID) http://orcid.org/0000-0001-6465-0094

### Decision letter and Author response

Decision letter https://doi.org/10.7554/eLife.31216.023
Author response https://doi.org/10.7554/eLife.31216.024

## Additional files

### Supplementary files

• Supplementary file 1. List of strains. The numbers and their corresponding genotypes are indicated for all strains used in this study.
DOI: https://doi.org/10.7554/eLife.31216.015

• Supplementary file 2. List of primers. All primers included in this study are identified by primer number, a brief description of how they were used and their corresponding sequence.
DOI: https://doi.org/10.7554/eLife.31216.016

• Supplementary file 3. List of plasmids. Plasmids used in this study are identified by number and accompanied by a brief description of how they were utilized.
DOI: https://doi.org/10.7554/eLife.31216.017

• Supplementary file 4. Chromosome rearrangement breakpoint analyses. A synopsis of the analyses we used to detect translocation breakpoints. Breakpoints were detected by LUMPY (*Layer et al., 2014*) and confirmed by PCR (for primer sequences, see *Supplementary file 2*).
DOI: https://doi.org/10.7554/eLife.31216.018

• Transparent reporting form
DOI: https://doi.org/10.7554/eLife.31216.019

### Major datasets

The following dataset was generated:

| Author(s) | Year | Dataset title | Dataset URL | Database, license, and accessibility information |
|---|---|---|---|---|
| Kirsty Jamieson, Kevin J McNaught, Tereza Ormsby, Neena A Leggett, Shinji Honda, Eric U Selker | 2017 | Telomere repeats induce domains of H3K27 methylation in Neurospora | https://www.ncbi.nlm.nih.gov/geo/query/acc.cgi?acc=GSE104019 | Publicly available at the NCBI Gene Expression Omnibus (accession no. GSE104019) |

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
