## [Decision Letter]

Thank you for submitting your article "Telomere repeats induce domains of H3K27 methylation in Neurospora" for consideration by *eLife*. Your article has been favorably evaluated by Kevin Struhl (Senior Editor) and three reviewers, one of whom is a member of our Board of Reviewing Editors. In light of the reviews, we invite you to submit a revised version of the manuscript.

Summary:

All reviewers agree that the manuscript is interesting, novel, and significant. However, some points need to be address for publication in *eLife*.

Essential revisions:

1) Please provide a better description of how ChIP is normalized.

2) A reviewer link to ChIP-seq dataset in GEO is also necessary for re-review.

3) What is the connection to PRC2? No experiments show that PRC2 is responsible for H3K27me3 at the newly integrated TTAGGG repeats. Is PRC2 recruitment required? Could TERRA be the connection or is there another method of recruitment?

4) Please show displays of data for the entire genome, not just the regions/chromosomes authors selected.

5) Please address the replicability of the experiments. The samples should not be pooled for analysis. Instead they should be sequenced and analyzed separately to determine how well replicated the results are for each.

6) A different targeted integration site (e.g., his3, pan2) would be a necessary control to rule out position effects.

---

## [Author Response]

Essential revisions:1) Please provide a better description of how ChIP is normalized.

We improved our description of the ChIP normalization in the Materials and methods.

2) A reviewer link to ChIP-seq dataset in GEO is also necessary for re-review.

The GEO accession is: GSE104019.

3) What is the connection to PRC2? No experiments show that PRC2 is responsible for H3K27me3 at the newly integrated TTAGGG repeats. Is PRC2 recruitment required? Could TERRA be the connection or is there another method of recruitment?

Attempts to ChIP PRC2 components in Neurospora have been unsuccessful but SET-7 is the only EZH2 homolog in Neurospora and we have shown that all naturally occurring H3K27me2/3 in Neurospora, including that which we now show is dependent on telomere sequences, is absolutely dependent on this protein (Jamieson et al. 2013). It seems exceedingly unlikely that another, unrelated enzyme catalyzes H3K27 methylation at the ectopic telomere repeats.

Our Discussion highlights both the possible involvement of TERRA and telomere-binding proteins in establishing H3K27me2/3 at ectopic TTAGGG repeats. Presuming TERRA is present in Neurospora, and transcribed from the ectopic TTAGGG repeats, it could act in *cis* by directly recruiting PRC2. PRC2 has a high affinity for the G-quadruplex structure that TERRA forms (Wang et al. 2017). Alternatively, PRC2 could be recruited to TTAGGG repeats by DNA binding factors. Recent experiments in Arabidopsis found that the telobox motif is enriched in candidate Polycomb response elements and that this motif can recruit the PRC2-interacting transcription factor, AZF1 (Xiao et al. 2017). Similar DNA binding factors in Neurospora may directly recruit PRC2 to the TTAGGG repeats.

4) Please show displays of data for the entire genome, not just the regions/chromosomes authors selected.

We now include this information (see new Figure 1—figure supplement 3 and Figure 5—figure supplement 2.).

5) Please address the replicability of the experiments. The samples should not be pooled for analysis. Instead they should be sequenced and analyzed separately to determine how well replicated the results are for each.

For all chromosomal rearrangement strains, H3K27me2/3 ChIP was performed in biological triplicate. Losses and gains of H3K27me2/3, essentially binary results, were confirmed by qPCR in biological triplicate. Some of our sequencing libraries were prepared using pooled triplicates to obtain the suggested minimum quantity of DNA required for our sequencing kit. In every experiment, qPCR results validated ChIP-seq results, giving us confidence in the ChIP-seq results.

We would like to emphasize that our paper describes dramatic, qualitative and highly specific changes in H3K27me2/3. Vast domains of H3K27me2/3 were lost and gained but only at regions moved from or to the vicinity of chromosome ends, respectively. The reproducibility of H3K27me2/3 distribution in regions unaffected by chromosomal rearrangements (see Figure 1—figure supplement 3) highlights the specificity of the observed H3K27me2/3 changes.

6) A different targeted integration site (e.g., his3, pan2) would be a necessary control to rule out position effects.

As suggested, we created an additional independent insertions of telomere repeats at a second locus, *his-3*, and demonstrated via ChIP-qPCR that 21, 22, and 23 telomere repeats also induce H3K27me (Figure 5—figure supplement 1.).